# Comaneci-Assisted Coiling of Wide-Necked Intracranial Aneurysm: A Single-Center Preliminary Experience

**DOI:** 10.3390/jcm11226650

**Published:** 2022-11-09

**Authors:** Gabriele Vinacci, Angelica Celentano, Edoardo Agosti, Alberto Vito Terrana, Francesco Alberto Vizzari, Luca Nativo, Fabio Baruzzi, Antonio Tabano, Davide Locatelli, Andrea Giorgianni

**Affiliations:** 1Neuroradiology Department, Circolo Hospital, ASST Sette Laghi, 21100 Varese, Italy; 2Department of Radiology, University of Insubria, 21100 Varese, Italy; 3Neurosurgery, Department of Medical and Surgical Specialties, Radiological Sciences and Public Health, University of Brescia, 25121 Brescia, Italy; 4Division of Neurosurgery, Department of Biotechnology and Life Sciences, University of Insubria, 21100 Varese, Italy

**Keywords:** Comaneci, assisted coiling, intracranial aneurysm, embolization

## Abstract

Background: Wide-necked aneurysms remain challenging for both coiling and microsurgical clipping. They often require additional techniques to prevent coil prolapse into the parent artery, such as balloon- and stent-assisted coiling. Comaneci is an expandable and removable stent that acts as a bridging device and does not interfere with the blood flow of the parent artery. Methods: We retrospectively reviewed our institutional radiological and clinical chart of patients treated for saccular intracranial aneurysm via endovascular Comaneci-assisted coiling. The aim of the study was to report our preliminary experience in Comaneci-assisted coiling of wide-necked intracranial aneurysms. Results: We included 14 patients in the study. Of these, 11 had a ruptured intracranial aneurysm and were treated with Comaneci-assisted coiling. We registered five minor intraprocedural complications and two intraprocedural failures of the device. At one-year follow-up, a satisfying aneurysm occlusion was observed in 85% of the cases. Conclusions: Though long-term follow-up data and larger case series are needed, this preliminary study showed the feasibility of the Comaneci-assisted coiling method for both ruptured and unruptured wide-neck intracranial aneurysms, with similar occlusion rates as balloon-assisted coiling. However, we registered high incidence of thromboembolic complications; these were probably related to the lack of heparin administration. The main advantageous application of this technique is likely in cases of ruptured intracranial aneurysms, as there is no need for post-procedural antiplatelet therapy.

## 1. Introduction

Though endovascular treatment of intracranial aneurysms has been widely performed for both ruptured and unruptured intracranial aneurysms [1,2,3] in support of and replacing surgical treatment, wide-necked aneurysms (WNAs) remain a challenging morphological type of aneurysm. 

Endovascular coiling for WNAs is associated with a high risk of coil mass protrusion into the parent vessel and consequent ischemia of downstream territories [4]. Indeed, in comparison with narrow-neck aneurysms, endovascular embolization of WNAs is associated with higher procedural risks and poorer long-term follow-up [4]. To avoid coil prolapse and major ischemic iatrogenic complications, WNAs usually require additional techniques, such as balloon-assisted coiling (BAC), stent-assisted coiling (SAC), or the application of a flow-diverter stent or emerging endovascular devices [5,6,7,8]. These additional techniques may increase the risk of thromboembolic complications during the procedure and, in certain cases, require dual antiplatelet therapy [9,10,11]. Both BAC and SAC are currently applied in the treatment of WNAs to protect the parent vessel in either a temporary or permanent fashion [12,13]. However, BAC temporarily obstructs flow; meanwhile, the use of stents necessitates dual antiplatelet therapy which increases both procedural and long-term bleeding risks [14,15].

We describe our early experience with a Comaneci bridging device in the treatment of both ruptured and unruptured WNAs. The main advantage of this novel technique is the possibility of performing a temporary bridging on neck aneurysms without blood flow occlusion; its intrinsic soft structure may reduce the risk of vessel rupture compared with remodeling balloons; finally, it does not require dual antiplatelet therapy (DAPT), because it is a self-expanding stent and is fully removable.

The aim of the study was to retrospectively describe our early experience with Comaneci bridging devices in the treatment of both ruptured and unruptured WNAs among 14 patients.

## 2. Methods

This is a monocentric study. We retrospectively collected the data for patients with WNAs who were treated for saccular intracranial aneurysm via endovascular Comaneci-assisted coiling. We collected data of the patients treated between October 2015 (our institution’s first case) and May 2021. Comaneci-assisted coiling was performed by two senior operators with 12 and 6 years of experience, respectively, at the time of the end of enrollment. The Comaneci (Rapid Medical, Yokneam, Israel) bridging device is a controllable, nondetachable, and retrievable temporary bridging device which was recently introduced to assist in the coiling process [16,17,18,19]. As a bridging device, the Comaneci device does not interfere with the blood flow of the parent artery and it can be expanded according to the aneurysm neck morphology and vessel requirement [17]. 

The Comaneci device consists of three device models: Comaneci, Comaneci Petit, and Comaneci 17. These are adaptable for vessel diameters of 1.5–4.5 mm, 1.5–3.5 mm, and 0.5–3 mm, respectively. The devices consist of a nitinol fine wire mesh region at the distal end, which is mounted on a flexible shaft that expands and contracts directly when the handle, with a control slider, is opened and closed. The wires of the mesh are radiopaque, which allows great visibility under fluoroscopy. The Comaneci has a flexible and soft distal tip, which allows safe navigation into the vessel.

WNAs have been defined as aneurysms with an absolute neck width >4 mm or a dome-to-neck ratio of <2 mm. All medical records and imaging were revised. The neurological assessments at discharge and at the 3-month follow-up were evaluated by a modified ranking scale (mRS). Magnetic resonance imaging was performed at the 3-month follow-up, and digital subtraction angiography was performed at the 1-year follow-up. The modified Raymond–Roy occlusion classification was used to classify the aneurysmal status. 

The primary goals of our retrospective data collection were to assess the safety of the Comaneci method, measured by the number of complications after the procedure, the efficacy of the Comaneci, measured by the required deployment of a stent or the need of microsurgical conversion, and the thromboembolic risk of the Comaneci, evaluated as the number of intraprocedural thromboembolic events. 

### 2.1. Antiplatelet and Antithrombotic Management

The protocol of antiplatelet and anticoagulant medications was variable depending on the SAH. For 5 days before endovascular embolization, 100 mg Acetylsalicylic acid and 75 mg/day Clopidogrel were provided for patients with unruptured aneurysms (in anticipation in case of treatment failure and the need for permanent stent placement). The DAPT was discontinued after the procedure if the Comaneci-assisted embolization was successful. Among the patients with ruptured aneurysms, no preventive antiplatelet therapy was provided. In both cohorts, according to our institution protocol, no intravenous heparin was administered, all the catheters were continuously perfused with saline infused with heparin (1500 UI/L), and no intravenous heparin was infused. In our institution, we continuously flush the catheters with heparin during aneurysm embolization; according to our experience, this practice reduces the risk of clot formation inside the catheters. 

### 2.2. Endovascular Procedure

All the procedures were performed through access to the right femoral artery. Through a 7F catheter placed in the ICA or the vertebral artery, two microcatheters were used: one 0.017″ microcatheter for aneurysm catheterization and one 0.021″ or 0.017″ microcatheter for the positioning of the Comaneci in the parent artery. 

In all cases, a three-dimensional rotational angiography (3D-DSA) was performed and the images were evaluated to determine the necessary sizes for the coils and the Comaneci device.

First, a 0.021″ microcatheter for the Comaneci or Comaneci Petit devices, or a 0.017″ microcatheter for the Comaneci 17 device, was navigated in the parent artery; then, a 0.017″ microcatheter was navigated in the aneurysm sack (Figure 1). The Comaneci device was then opened until a satisfying neck bridging was obtained; the coil was then positioned in the sack (Figure 1). Before the detachment of every coil, the Comaneci device was closed to evaluate the stability and the correct intrasaccular disposition of the coils, as well as to exclude the embedding of the coils through the Comaneci device. A post-procedural CT was performed in every patient to determine the occurrence of any periprocedural complications.

## 3. Results

### 3.1. Population

We included 14 patients diagnosed with saccular intracranial aneurysms who received Comaneci-assisted endovascular coiling. Nine patients (64%) were female and the mean age was 62.3 years (range 45–86 years) (Table 1). 

Eleven (79%) patients were admitted with subarachnoid hemorrhage (SAH), one (7%) patient had a persistent headache, one (7%) patient had a post-clipping growing neck remnant, and one (7%) patient was diagnosed during magnetic resonance imaging (MRI) screening for familiarity. Seven aneurysms (50%) were in the anterior communicating artery (ACom), five aneurysms (36%) were in the communicating internal carotid artery (ICA), one aneurysm (7%) was in the anterior medullary segment of the posteroinferior cerebellar artery (PICA), and one aneurysm (7%) was in the middle cerebral artery (MCA). The mean aneurysm height was 5.83 mm (range 2–14.5 mm), and the mean maximal width was 4.33 mm (range 2–15.6 mm), with mean neck size of 3.14 mm (range 2–8.2 mm). 

We deployed four (29%) standard Comaneci devices, eight (57%) Comaneci 17 devices, and two (14%) Comaneci Petit devices. No navigation or deployment problems were observed. In 12 (86%) patients, the Comaneci device was able to be deployed to successfully bridge the aneurysmal neck; in 2 (14%) patients, the Comaneci device did not achieve a satisfying scaffolding effect on the neck of the aneurysm and a self-expandable stent was required. 

Modified Raymond–Roy occlusion classification (Table 1) at the final angiographic images determined Class I in six (43%) cases, Class II in five (36%) cases, and Class IIIb in two (14%) cases. No migration of the Comaneci device was observed during its expansion or vasospasm in the parent artery. In all cases, an excellent visualization of the device was observed. 

### 3.2. Complications

Intraprocedural complication at least was observed in 5/14 patients (35%) (Table 1). In 3/14 (21%) cases, we observed clot formation inside the mesh of the Comaneci device (Figure 2), which rapidly regressed with the administration of 500 mg of Acetylsalicylic acid and Tirofiban (dose by weight; bolus infusion in 30 min followed by 12 h infusion). In 2/14 (14%) cases, after the infusion of 500 mg Acetylsalicylic acid and Tirofiban, the deployment of a rescue Atlas stent (Stryker Neurovascular, Salt Lake City, UT, USA) (Figure 2) was necessarily followed by 1 month of DAPT and a subsequent 3 months of Acetylsalicylic acid 100 mg/die. In one of the cases where a rescue Atlas stent was released, we had a failure of the Comaneci device, which was not able to perform a sufficient scaffold on the aneurysm neck; during the attempt of embolization, we observed intra-Comaneci clot formation, and an SAC was immediately performed. In the other case, the Comaneci-assisted embolization was performed but coils protruded on the aneurysm neck and intra-Comaneci clot formation was observed; then, we released an Atlas stent to ensure parent vessel patency. Notably, all the complications reported occurred in ruptured aneurysms in which no antiplatelet drugs were administered before the embolization. 

In all cases, no ischemia was observed at follow-up imaging, documented by strict CT follow-up for at least 30 days after the embolization. No intraprocedural bleeding or delayed aneurysm bleeding was observed. 

### 3.3. Follow-Up

In a range of 12–18 months, DSA follow-ups were performed; of the patients, 10 (71%) had a complete aneurysm occlusion, 2 (14%) had a small neck remnant that did not require any further endovascular or microsurgical intervention, and 1 (7%) had revascularization of the aneurysm sack and required the deployment of a flow diverter stent (FDs). Of 14 patients, 13 (93%) underwent CTs at 3 and 6 months, that demonstrated no delayed aneurysm bleeding or the appearance of acute or chronic ischemic lesions. Clinical status, assessed by mRs at 3 months, showed good outcome (mRs 0–2) in 11 (79%) patients, and poor outcome (5–6) in 3 (21%) patients (Table 1). Among the patients with poor outcome, two patients had a ruptured intracranial aneurysm and developed severe intracranial vasospasm, and one patient had a pretreatment of 5 mRs, and no worsening of mRs was observed.

## 4. Discussion

Despite our preliminary experience, our results show the feasible use of Comaneci device as a temporary bridging device in the coil-assisted treatment of WNAs. Since its approval, the Comaneci device has been implemented in different modalities and different cerebrovascular pathologies [20,21,22].

Comaneci-assisted embolization ranks between BAC and SAC. Both for BAC and SAC, the aim is to assist the coils’ deployment while preventing coils’ herniation into the parent vessel [23,24]. BAC involves the temporary vessel occlusion by the inflation of the balloon across the neck of an aneurysm, while SAC involves the placement of a self-expanding stent to support the coils across an aneurysm neck [24]. Although both methods have been demonstrated to be safe and efficient, BAC may be preferable in some clinical situations because it does not routinely require antiplatelet therapy [23]; however, in tortuous anatomy, the navigability of the balloon may be limited. Comaneci-assisted coiling possessed the characteristics of both BAC and SAC: it has the advantage of performing a temporary bridging effect: the balloon avoids the flow occlusion because it is a self-expanding stent. These advantages may induce a lower risk of thromboembolic events; in addition, its intrinsic soft structure may reduce the risk of vessel rupture during assisted embolization compared with balloons that present a stiffer structure. However, a potential limitation of the Comaneci device is the impossibility of blocking the blood flow in the event of an intraprocedural rupture. Although Comaneci device do not require antiplatelet therapy, its tendency to include clot formation has been described. Sikarov et al. [25] demonstrated an incidence of 5.93% intra-Comaneci clot formation; in their serie the clots rapidly regressed with abciximab infusion. Molina-Neuvo et al. [26] and Fisher et al. [17] observed 1.7% and 7.1% clot formation likelihood, respectively. In our study, we observed a 35% likelihood of clot formation. The high incidence we observed, compared with the previous studies [16,18,19,25], may be related to the lack of intravenous heparin administration. We observed thrombotic complications only in the group of ruptured aneurysms, in which neither antiplatelet nor anticoagulant were administered; meanwhile, in the unruptured group, in which DAPT was previously administrated, no thromboembolic complication occurred. We believe that intravenous heparin administration may reduce the incidence of thrombotic complications as it may reduce platelet activation in the Comaneci device mesh. Although we observed thrombotic complications in five patients, which rapidly regressed after medical therapy, no ischemic lesion was observed during the 1-month follow-up strict CT and clinical follow-up except for one patient, where severe and unresponsive diffuse intracranial vasospasm occurred. Although we were not able to perform an MRI, and we might have underestimated the ischemic lesions, none of the patients reported any symptoms related to the complications. While the risks of embedding the coil through Comaneci meshes have been noted [18,26], we did not observe them in any procedure. This risk may be reduced by closing the Comaneci device before the detachment of every coil and, if a protrusion of the coils is observed, the recovery of the coil and a new deployment is mandatory. We observed a satisfying occlusion rate of 86% at follow-up. This rate is similar to that of BAC but lower than that of SAC [27,28,29]. However, in our series, we observed a higher rate of complications than those that occur with BAC and SAC [27,28,29]; this may be related to our early experience and the small cohort. Compared with our experience with other embolization techniques in ruptured WNAs, such as SAC and FDs, we observed a higher incidence of thromboembolic complications using Comaneci-assisted coiling. In our experience the incidence of thromboembolic complications with SAC is 6% and that with FDs is 11%; however, the rebleeding rates were higher in both groups (8% and 11%, respectively). Compared with the aforementioned techniques, we observed a higher incidence of thromboembolic complications with the Comaneci device, but we did not observe any rebleeding. The real advantage of the Comaneci device compared with a self-expandable stent is the fact that antiplatelet therapy is not necessary, especially in ruptured scenario, the absence of antiplatelet therapy may reduce the severity of a possible rebleeding and would allow the safer clinical and neurosurgical management of SAH. 

In agreement with the results of this study, Comaneci-assisted coiling is a feasible technique in cases of wide-necked intracranial aneurysms. In our study, we found that the safety of the procedure was unclear, given the high incidence of thrombotic complications. However, we believe that, in order to avoid and reduce this risk (in accordance with previous studies), anticoagulation is mandatory, especially for ruptured aneurysms. However, studies with a greater number of cases are needed to support the preliminary evidence that emerged in this study and to more clearly define the possible periprocedural complications. The best application of this technique is likely in cases of ruptured intracranial aneurysms, since there is no need for post-procedural antiplatelet therapy.

### Limitations

The main limitations of this study are its retrospective nature, with a small number of patients and the lack of a long-term follow-up. Furthermore, the absence of a BAC control group limits our ability to extrapolate our findings on the efficacy of Comaneci-assisted coiling.

## Figures and Tables

**Figure 1 jcm-11-06650-f001:**
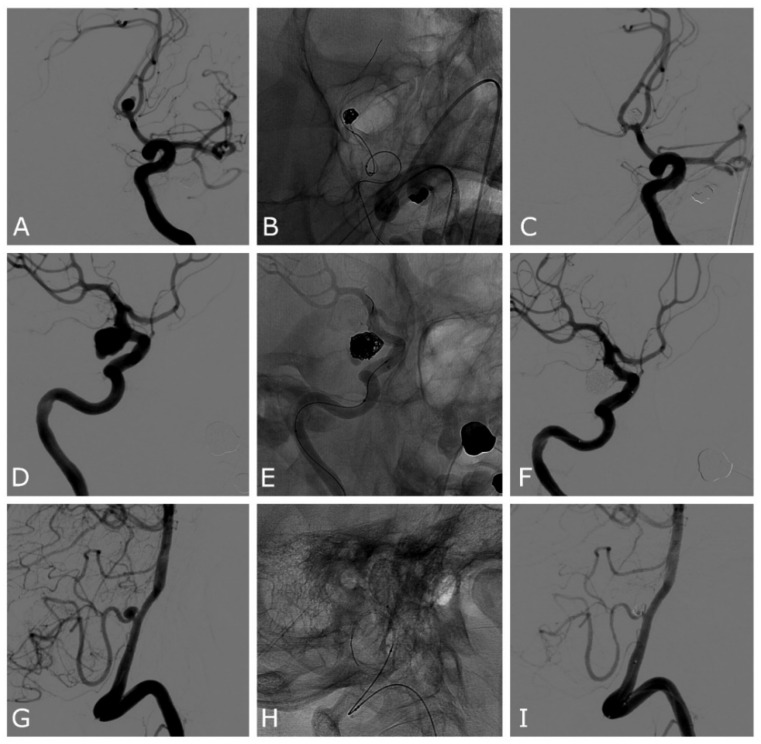
Patient number 13 (**A**–**C**): ruptured ACOM wide-neck aneurysm. (**A**) Anteroposterior oblique 2D DSA demonstrating the ACOM aneurysm with a daughter sac pointing superiorly. (**B**) Unsubtracted image showing the expanded Comaneci device after the first framing coil is deployed. (**C**) DSA final result demonstrating complete occlusion of the aneurysm and patency of both the ACAs. Patient number 5 (**D**–**F**): unruptured PCOM wide-neck aneurysm. (**D**) Lateral oblique 2D DSA demonstrating the PCOM aneurysm. (**E**) Unsubtracted image showing the expanded Comaneci device after the first framing coil is deployed. (**F**) DSA final result demonstrating near-complete occlusion of the aneurysm, with a small neck remnant in the inferior part and patency of ICA. Patient number 3 (**G**–**I**): ruptured PICA wide-neck aneurysm. (**G**) Lateral oblique 2D DSA demonstrating the PICA aneurysm. (**H**) Unsubtracted image showing the expanded Comaneci device and the 0.017″ microcatheter in the aneurysm sack. (**I**) DSA final result demonstrating complete occlusion of the aneurysm, with a small neck remnant in the inferior part and patency of PICA. DSA—digital subtracted angiography; PCOM—posterior communicating artery; ICA—internal carotid artery; ACA—anterior cerebral artery; ACOM—anterior communicating artery; PICA—posterior inferior cerebellar artery.

**Figure 2 jcm-11-06650-f002:**
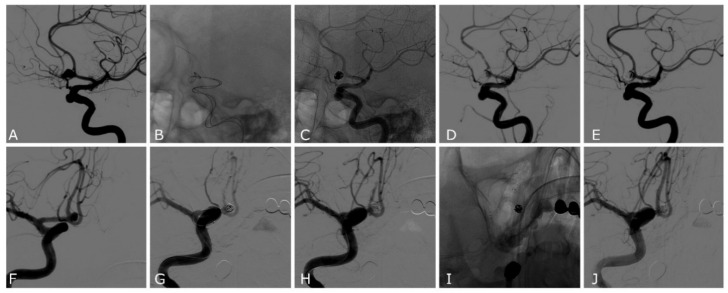
Patient number 11 (**A**–**E**): ruptured ACOM wide-neck aneurysm. (**A**) Lateral oblique 2D DSA demonstrating the ACOM aneurysm with a daughter sac pointing anteriorly. (**B**,**C**) Unsubtracted image showing the expanded Comaneci device during the coiling. (**D**) Lateral oblique 2D DSA showing the clot formation between the meshes of the Comaneci device that required the infusion of Acetylsalicylic acid and Tirofiban. (**E**) Lateral oblique 2D DSA acquired 30 min after the infusion of Acetylsalicylic acid and Tirofiban demonstrating a complete resolution of the thrombotic complication and a neck remnant of the aneurysm. Patient number 12 (**F**–**J**): ruptured ACOM wide-neck aneurysm. (**F**) Anteroposterior oblique 2D DSA demonstrating the ACOM aneurysm. (**G**) Anteroposterior oblique 2D DSA demonstrating clot formation between the meshes of Comaneci device that required infusion of Acetylsalicylic acid and Tirofiban. (**H**) Anteroposterior oblique 2D DSA after removing the Comaneci device, showing the protrusion of the coils in the parent artery and thromboembolic occlusion of the right A2 segment of anterior cerebral artery, which was not yet responsive to medical treatment. (**I**) Unsubtracted image of the deployment of a rescue Atlas stent. (**J**) Anteroposterior oblique 2D DSA acquired 30 min after the infusion of Acetylsalicylic acid and Tirofiban, demonstrating the patency of the Atlas stent, the resolution of the clot, and the patency of both anterior cerebral arteries.

**Table 1 jcm-11-06650-t001:** Patients ordered chronologically. Demographic characteristics of the patients and aneurysm characteristics, device used, and operative outcome are displayed. ACOM—anterior communicating artery; PICA—posterior inferior cerebellar artery; MCA—middle cerebral artery; PCOM—posterior communicating artery.

Case	Age	Sex	Location	SAH	Max Size (Height, Width, Neck) mm	Device	Complications	Rescue Treatment	mRR	mRs at 3 Months
**1**	74	F	PCOM	No	4.2, 7, 4	Comaneci	-		I	0
**2**	65	M	PCOM	Yes	3.1, 4, 3	17	-		I	0
**3**	70	F	PICA	Yes	3.5, 2, 1.5	Comaneci	-		I	0
**4**	86	F	PCOM	Yes	6.4, 4.3, 5.5	Comaneci	Intra-Comaneci and parent artery clot formation	Stent	II	6
**5**	59	F	ACOM	Yes	10, 6.2, 5.5	17	-		II	0
**6**	65	F	PCOM	No	6.7, 3.9, 3	Petit	-		II	0
**7**	51	F	MCA	No	2, 2.2, 2	Petit	-		II	5
**8**	65	M	ACOM	Yes	5.7, 3.7, 3.5	17	-			0
**9**	77	F	PCOM	Yes	8, 10.3, 5.9	Comaneci	Intra-Comaneci clot formation		I	0
**10**	65	F	ACOM	Yes	3, 2.4, 2	17	-		I	5
**11**	45	M	ACOM	Yes	5.5, 3.6, 2.8	17	Intra-Comaneci clot formation		IIIb	1
**12**	62	F	ACOM	Yes	9, 4.5, 3	17	Intra-Comaneci clot formation	Stent	IIIb	0
**13**	55	M	ACOM	Yes	14.5, 15.6, 8.2	17	-		I	0
**14**	61	M	ACOM	Yes	3, 5.9, 5	17	Intra-Comaneci clot formation		II	0

## Data Availability

No new data were created or analyzed in this study. Data sharing is not applicable to this article.

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
