# Peer review of "Comaneci-Assisted Coiling of Wide-Necked Intracranial Aneurysm: A Single-Center Preliminary Experience"

_jcm, 2022, doi:10.3390/jcm11226650_

Round 1

Reviewer 1 Report

The presented subject is very interested and further researches should be counducted.

As it was already mentioned, despite the small study group, the method presented in the manuscript is important not only for neuroradiologists, but also for neurosurgeons. The angiographic examinations results presented in the article show a very good effect of the treatment. Due to the location of the treated aneurysms, the used method seems to be the safest and deserves attention and further dissemination.

Author Response

We thank the reviewer for this comment. As we had mentioned we reported our preliminary data and we confirmed the feasibly of the tecnique.

Reviewer 2 Report

The authors described their experience with the comaneci embolization device in 14 patients with wide-necked aneurysms.

1.  The number of patients included in the study is relatively small compared to other studies that have been published in the last five years.

2. In 3/14 patients the procedure was not completed with the comaneci device.

3. In 5/14 or in 5/11? patients intraprocedural complications were observed.

The above mentioned data are not consistent with the author's conclusion that Comaneci-assisted coiling is feasible and safe technique for the embolization of wide-necked cerebral aneurysms.

Author Response

Question 1: The number of patients included in the study is relatively small compared to other studies that have been published in the last five years.

We thank the reviewer for this comment. We reported our preliminary experience that however showed a higher incidence of thromboembolic complication, which we analyzed and related to the lack of intravenous Heparin administration. This may be helpful to underline the importance of Heparin administration even in a ruptured aneurysm scenario.

Question 2. In 3/14 patients the procedure was not completed with the comaneci device. 

We thank the reviewer for this comment. We modified the text and made it clearer. We had 2 insufficient scaffolding by Comaneci that required the deployment of a stent.

Question 3. In 5/14 or in 5/11? patients intraprocedural complications were observed.

We thank the reviewer for this comment. We modified the text and made it clearer. 5/14 complication, 3 that required Acetylsalicylic acid and Tirofiban and 2 that required Acetylsalicylic acid and Tirofiban and the deployment of a stent.

Question 4: The above mentioned data are not consistent with the author's conclusion that Comaneci-assisted coiling is feasible and safe technique for the embolization of wide-necked cerebral aneurysms.

We thank the reviewer for this comment. We modified the text. We believe that, in our study, the feasibility of Comaneci is in line with the previous study data. We had a higher incidence of thromboembolic complication, however, we believe that this data may be related to the lack of systemic heparin. We modified the text as we can not assume that the device is safe from our data, but we specified that the data may be negatively influenced by the lack of intravenous heparin.

Reviewer 3 Report

The authors report on their series of aneurysm patients treated with a newer endovascular neck remodelling device. This is a needed paper in the literature to help document the positives and negatives of using a new device.  My comments on the paper are below,

1: Methods:

More detail is required.  Is this a single centre study? How many operators were involved in treating the study patients? How experienced were these operators? Was the use of the comaneci stent used at this centre prior to the start of this trial or does this study include the entire centre's experience with this new device? If this study includes all patients treated with this device in their institution, when in the series did the described complications occur.

Regarding follow up, more detail is required to understand what data points were assessed.  Specifically in relation to imaging follow up. 

2: Results:

Antiplatelet management:

ASA and clopidogrel route and frequency are required.  Was a single dose or multiple doses administered.  ASA 100mg and Clopidogrel 75mg would not be considered a loading dose of these medications.  A rationale for the doses used should be included.  

Procedural heparinization: 

It is stated that heparin is not utilized for the endovascular treatment of aneurysm patients.  As this practice is variable among institutions, a statement regarding the rationale for this practice should be included. 

Complications:

New clot formation was successfully treated by ASA and tirofiban. Doses and routes of these medications need to be provided.

Details regarding imaging follow up and reporting of imaging related complications is poor.  It is stated that no ischemia was noted on follow up imaging.  However there is no description of how this was assessed on an MRI at 3 months post op.  There is no mention of any post op imaging prior to the 3 month MRI.  A detailed description of how post op imaging was assessed and what would have been considered evidence of post op ischemia should be included.  Similarly, how was the presence of delayed aneurysm bleeding assessed if there was no short interval post operative imaging performed or at least reviewed?

3: Discussion:

The authors report an extremely high incidence of thrombus formation on the Comaneci stent and/or coils (35%), which is at least 4 times higher than the references cited as a comparison. This required the administration of emergency medications of ASA and tirofiban. This should be considered a major adverse event. Despite this high risk, the authors conclude that Comaneci is a safe device as no cerebral ischemic lesions developed in the treated patients.  However, the authors provide no evidence to support this conclusion.  There is no post op imaging performed in the first week following treatment reported.  For example, an MRI with DWI could prove the presence or absence of a DWI lesion. While immediate post op MRI is not necessarily standard practice in most institutions, it is well documented that many endovascular procedures cause one or more small and often clinically silent DWI lesions following an uncomplicated procedure. The concern therefore is that in the setting of documented thrombus formation requiring emergency treatment, that the incidence of DWI lesions could be higher.  However an MRI at 3 months offers a limited assessment to support this statement of “no ischemic lesions”.  As an alternative, more details around the clinical examination of the patient at the time of discharge to support a statement of no clinically observable ischemic event, would be something to consider.

Additionally, given the high incidence of thrombotic complications in this series, the authors should offer some discussion as to why they feel this may have occurred.  One point of discussion not offered, is their institutional protocol of not using a heparin bolus or infusion during the endovascular management of aneurysms.  This practice is variable and the authors should discuss their rationale specifically regarding the risks and benefits of heparin use using the Comaneci stent.

Lastly, the identified limitation of this study having a lack of a control group is significant. The authors recognize this limitation, however it seems like a lost opportunity to not use their own patient series of wide necked aneurysms treated with either balloon or stent neck remodelling during the same treatment period in their institution. This addition would significantly strengthen the paper and it’s relevance to the treatment of intracranial aneurysms.     

Author Response

Question 1: More detail is required.  Is this a single centre study? How many operators were involved in treating the study patients? How experienced were these operators? Was the use of the comaneci stent used at this centre prior to the start of this trial or does this study include the entire centre's experience with this new device? If this study includes all patients treated with this device in their institution, when in the series did the described complications occur.

Regarding follow up, more detail is required to understand what data points were assessed.  Specifically in relation to imaging follow up

We thank the reviewer for this comment. We modified the text and reported the missing data.

Question 2:

Antiplatelet management:

ASA and clopidogrel route and frequency are required.  Was a single dose or multiple doses administered.  ASA 100mg and Clopidogrel 75mg would not be considered a loading dose of these medications.  A rationale for the doses used should be included.  

Procedural heparinization: 

It is stated that heparin is not utilized for the endovascular treatment of aneurysm patients.  As this practice is variable among institutions, a statement regarding the rationale for this practice should be included. 

Complications:

New clot formation was successfully treated by ASA and tirofiban. Doses and routes of these medications need to be provided.

Details regarding imaging follow up and reporting of imaging related complications is poor.  It is stated that no ischemia was noted on follow up imaging.  However there is no description of how this was assessed on an MRI at 3 months post op.  There is no mention of any post op imaging prior to the 3 month MRI.  A detailed description of how post op imaging was assessed and what would have been considered evidence of post op ischemia should be included.  Similarly, how was the presence of delayed aneurysm bleeding assessed if there was no short interval post operative imaging performed or at least reviewed?

We thank the reviewer for this comment. We modified the text and reported the missing data. The missing data remains the MRI follow-up. We were not able to perform an MRI either in the post-op or at 3 months. This is a limitation of the study because, as the reviewer underlined, CT underestimates the ischemic sequelae. We modified the text and we underlined that our assumption of no ischemic event in the discussion may be influenced by the lack of MRI follow-up. 

Question 3:

The authors report an extremely high incidence of thrombus formation on the Comaneci stent and/or coils (35%), which is at least 4 times higher than the references cited as a comparison. This required the administration of emergency medications of ASA and tirofiban. This should be considered a major adverse event. Despite this high risk, the authors conclude that Comaneci is a safe device as no cerebral ischemic lesions developed in the treated patients.  However, the authors provide no evidence to support this conclusion.  There is no post op imaging performed in the first week following treatment reported.  For example, an MRI with DWI could prove the presence or absence of a DWI lesion. While immediate post op MRI is not necessarily standard practice in most institutions, it is well documented that many endovascular procedures cause one or more small and often clinically silent DWI lesions following an uncomplicated procedure. The concern therefore is that in the setting of documented thrombus formation requiring emergency treatment, that the incidence of DWI lesions could be higher.  However an MRI at 3 months offers a limited assessment to support this statement of “no ischemic lesions”.  As an alternative, more details around the clinical examination of the patient at the time of discharge to support a statement of no clinically observable ischemic event, would be something to consider.

Additionally, given the high incidence of thrombotic complications in this series, the authors should offer some discussion as to why they feel this may have occurred.  One point of discussion not offered, is their institutional protocol of not using a heparin bolus or infusion during the endovascular management of aneurysms.  This practice is variable and the authors should discuss their rationale specifically regarding the risks and benefits of heparin use using the Comaneci stent.

Lastly, the identified limitation of this study having a lack of a control group is significant. The authors recognize this limitation, however it seems like a lost opportunity to not use their own patient series of wide necked aneurysms treated with either balloon or stent neck remodelling during the same treatment period in their institution. This addition would significantly strengthen the paper and it’s relevance to the treatment of intracranial aneurysms.     

We thank the reviewer for this comment. We modified the text and we excluded the safety of the device. As the author reported we had a higher incidence of thromboembolic complications compared to the literature, and we focused on the lack of intravenous heparin administration. The lack of intravenous heparin administration may have influenced the safety of the device. However the patients, except one that died because of severe and untreatable intracranial vasospasm, did not present any focal neurologic deficit. Although the overlapping of possible SAH sequelae may be a confounding factor. As the author suggested we included in the discussion our data regarding the other embolization techniques for intracranial wide neck aneurysms. We had a higher thromboembolic complication with Comaneci but a lower incidence of rebleeding. However, we can not include balloon-assisted coiling in this analysis as we do not perform it in our institution.